# Ectopic Mediastinal Thyroid: A Crossroad Between a Multi-Layered Endocrine Perspective and a Contemporary Approach in Thoracic Surgery

**DOI:** 10.3390/life14111374

**Published:** 2024-10-25

**Authors:** Claudiu Nistor, Mihai-Lucian Ciobica, Oana-Claudia Sima, Anca-Pati Cucu, Florina Vasilescu, Lucian-George Eftimie, Dana Terzea, Mihai Costachescu, Adrian Ciuche, Mara Carsote

**Affiliations:** 1Department 4—Cardio-Thoracic Pathology, Thoracic Surgery II Discipline, “Carol Davila” University of Medicine and Pharmacy, 0505474 Bucharest, Romania; claudiu.nistor@umfcd.ro (C.N.); adrian.ciuche@umfcd.ro (A.C.); 2Thoracic Surgery Department, “Dr. Carol Davila” Central Emergency University Military Hospital, 010825 Bucharest, Romania; anca-pati.cucu@drd.umfcd.ro (A.-P.C.); mihaicostachescu@gmail.com (M.C.); 3Department of Internal Medicine and Gastroenterology, “Carol Davila” University of Medicine and Pharmacy, 020021 Bucharest, Romania; 4Department of Internal Medicine I and Rheumatology, “Dr. Carol Davila” Central Military University Emergency Hospital, 010825 Bucharest, Romania; 5Department of Clinical Endocrinology V, C.I. Parhon National Institute of Endocrinology, 011863 Bucharest, Romania; carsote_m@hotmail.com; 6PhD Doctoral School of “Carol Davila” University of Medicine and Pharmacy, 0505474 Bucharest, Romania; 7Department of Pathology, “Dr. Carol Davila” Central Military University Emergency Hospital, 010825 Bucharest, Romania; florinapath@yahoo.com (F.V.); lucian.eftimie@unefs.ro (L.-G.E.); 8Discipline of Anatomy and Biomechanics, National University of Physical Education and Sports, 060057 Bucharest, Romania; 9Department of Pathology, C.I. Parhon National Institute of Endocrinology, 011863 Bucharest, Romania; danaterzea@gmail.com; 10Department of Endocrinology, “Carol Davila” University of Medicine and Pharmacy, 0505474 Bucharest, Romania

**Keywords:** minimally invasive thoracic surgery, cervicotomy, ectopic, mediastinum, dyspnoea, adrenal, COVID-19, Cooper retractor, thymus, thyroid

## Abstract

An ectopic thyroid (ET) involves numerous scenarios of detection and outcomes, while its current management is not standardised. A mediastinal ET (MET) represents a low index of suspicion. In this paper, we introduce a 47-year-old female who was accidentally identified with an MET, and a modern surgical approach was provided. An anterior mediastinal mass of 3.2 cm was found at CT upon a prior COVID-19 infection. Previous to the infection, she experienced non-specific complaints for a few months (intermittent night sweats, facial erythema, chest pressure, and dyspnoea). Also, CT identified a thymus-like mass and a left adrenal incidentaloma of 3 cm. The endocrine panel was normal, and the subject declined further investigations. She was re-admitted 12 months later: the MET had increased +1 cm (+45% volume) and was confirmed at a 99mTc pertechnetate scintigraphy. Noting the symptoms, mediastinal anatomy, and size change, the MET was removed via a minimally invasive trans-cervical approach (eutopic gland preservation) with the help of a Cooper thymectomy retractor (which also allowed for a synchronous thymus mass resection). No post-operatory complications were registered, the thyroid function remained normal, and the mentioned symptoms were remitted. A histological exam confirmed a benign MET and thymus hyperplasia, respectively. To conclude, this case pinpoints important aspects, such as the clinical picture became clear only upon thoracic surgery due to the complete remission of the complaints that initially seemed widely non-specific. The incidental MET finding was associated with a second (adrenal) incidentaloma, a scenario that might not be so rare, following multiple imaging scans amid the COVID-19 era (no common pathogenic traits have been identified so far). The co-presence of a thymus mass represented one more argument for surgery. Minimally invasive cervicotomy associated with eutopic gland conservation and the use of a Cooper thymectomy retractor highlight modern aspects in video-assisted thoracic surgery, which provided an excellent outcome, involving one of the lowest mediastinal thyroids to be removed by this specific procedure. Awareness of such unusual entities helps inform individualised, multidisciplinary decisions for optimum prognoses.

## 1. Introduction

Ectopic thyroid tissue involves numerous scenarios of detection and outcomes, depending on its site, size, compressive elements, histological profile, co-morbidities (for instance, prior or concurrent cancers), and specific management, for which there is not currently a standardised (guideline-based) approach [1,2,3]. A mediastinal thyroid may mimic many other benign and malignant non-endocrine masses or ectopic parathyroid tumours; it represents a low case index of suspicion from an epidemiological perspective [4,5,6].

When analysing retrosternal ectopic thyroid tissue (which is distinct from retrosternal goitre), the main features that should be taken into account include, firstly, its clinical presentation, which is non-specific, and may be completely absent [7,8,9], followed by standard thyroid hormones, autoantibodies, and tumour marker assays [10,11]. The imaging exam of an orthotopic thyroid essentially includes thoracic computed tomography, while eutopic thyroid should be initially assessed via an anterior neck ultrasound assessment [12,13,14].

One particular trait involves the identification of the connective tissue with the cervical gland, which may be achieved based on imaging scans, intra-operatory findings, or a post-surgery histological exam, and it helps make the distinction between an accessory thyroid lobe and a retrosternal/substernal goitre [15,16,17]. An important aspect is represented by the careful investigation of suspected thyroid malignancy in a mediastinal and/or cervical thyroid, which represents a milestone in multidisciplinary management [18,19,20].

Overall, the best strategy in terms of interventional techniques for diagnosis and tissue removal is based on three main pillars:The identification of a specific functional pattern in the ectopic thyroid, using, for example, I^131^ radioiodine or technetium (99m-Tc) scintigraphy;The decision to directly access the ectopic thyroid via fine needle aspiration or biopsy (if feasible and not considered too risky in individuals, where it will help the overall prognosis or in subjects who are not surgery candidates) [21,22,23];The surgical team strategy is to remove the ectopic tissue. A surgical (rather than conservative) approach is currently considered the best option in most mediastinal cases, and minimally invasive access is generally regarded as safe in the modern era of thoracic surgery, but it requires an individualised decision and an experienced team [24,25,26,27,28,29,30,31,32].

## 2. Objective

We aim to introduce a complex case study of an adult female accidentally identified with a benign ectopic mediastinal thyroid. A contemporary surgical approach was provided in terms of minimally invasive video-assisted cervicotomy with eutopic thyroid preservation and access via a Cooper thymectomy retractor (Figure A3).

Other interesting facts are its detection amid the surveillance of a prior COVID-19 infection, the co-presence of an adrenal incidentaloma as a second endocrine anomaly, and a tumour-like mass at the thymus level.

## 3. Methods

In addition to the case presentation, a brief literature review was performed with respect to the specific highlights of this paper. In the Discussion section, we mention our PubMed search of full-length, English language, original papers from its inception until 22 July 2024, which looked for similar reports focusing on the main unusual findings and specific surgical management (e.g., cervicotomy as a type of incision for removing ectopic thyroid tissue within the upper mediastinum, associated thymectomy, and the use of a Cooper retractor).

### 3.1. Case Presentation

#### 3.1.1. Baseline Admission: Identification of a Mediastinal Mass and an Adrenal Tumour Following Prior COVID-19 Infection

This patient was a female 47-year-old non-smoker admitted for an evaluation after she underwent a mild COVID-19 infection eight months prior (she did not require hospitalisation at that time). She experienced, for a few months, mild, non-specific complaints previous to the coronavirus infection, such as intermittent night sweats, facial erythema, mild chest pressure, and dyspnoea during moderate physical effort. The subject was a healthcare worker at a tertiary centre of endocrinology. Her personal and family medical history was irrelevant. Other than a body mass index of 30 kg/sqm, the physical exam was within normal limits, and she had regular menses.

Contrast-enhanced computed tomography revealed mild bilateral bronchiectasis and fine fibrotic sequels in the basal pyramid of the lower pulmonary lobes. Also, the scan identified a dense, heterogeneous, iodophile nodule with micro- and macro-calcifications in the anterior upper mediastinum, which caudal-laterally displayed bilateral contact with the subclavian arteries. (The mass was situated within the lowest part of the upper mediastinum, close to the middle mediastinum.) Moreover, a computed tomography scan pinpointed a hypodense nodule (with a tumour-like appearance) at the level of the thymus area (suspected of being a thymus remnant), and the left adrenal gland presented an incidentaloma, suggestive of an adrenal cortical adenoma (Figure 1).

Under these circumstances, the blood assays identified a slightly elevated level of liver enzymes (of note, she had negative tests for viral hepatitis B and C). Normal thyroid function and negative thyroid autoimmunity were confirmed, as well as a normal adrenal hormone profile (Table A1). The patient was further referred for a second opinion to pneumology and thoracic surgery departments with regard to the mediastinal masses, which she declined, and she returned for re-assessment after one year.

#### 3.1.2. Endocrine and Imaging Investigations After One Year of Surveillance

The biochemical and endocrine profiles remained normal (Table A1). The thyroid ultrasound aspects were stationary (Figure A4). A computed tomography scan re-confirmed a dense tumour mass in the anterior mediastinum, with an increased size versus the prior one-year scan, extending 2.34 cm below the jugular notch. The initial volume of the mediastinal mass registered a +45.61% increase from 11.63 cm^3^ to 16.93 cm^3^. A native computed tomography scan confirmed a hypodense nodule at the level of the thymus area. An abdominal scan re-confirmed a stationary left adrenal incidentaloma (Figure 2).

The adrenal incidentaloma remained non-secreting (Table A1), and no surgical decision was taken at that time, but long-term surveillance was recommended.

#### 3.1.3. Ectopic Mediastinal Thyroid Tissue: 99m-Tc Scintigraphy Exploration

This time, the patient agreed to carry on with a multidisciplinary team evaluation, and, upon suspicion of an ectopic mediastinal thyroid, which was raised by the thoracic surgery unit, a further 99m-Tc pertechnetate scintigraphy was performed. The investigation revealed a low-positioned cervical thyroid with a diffusely inhomogeneous tracer uptake and an inhomogeneous area of tracer uptake at the level of the median line in the upper mediastinum, apparently without contact with the neck thyroid, hence suggesting an ectopic upper mediastinal thyroid (Figure 3).

### 3.2. Ectopic Mediastinal Thyroid Removal via Cervicotomy

No malignancy was suspected with concern to the eutopic thyroid; thus, no further fine needle aspiration was performed. The removal of the ectopic tissue was considered essential due to its anatomical position, its potential to develop more severe compressive symptoms, and the fact that the one-year conservative approach under imaging surveillance showed a 1 cm increase in its diameter.

She underwent surgical removal of the ectopic mediastinal mass and the rest of the thymic mass through minimally video-assisted invasive cervicotomy. The thoracic surgery team used a Cooper thymectomy retractor (Figure A1), which was inserted with the traction blade placed retrosternal (Figure 4).

The cervical thyroid was macroscopically normal. During the mediastinal surgical exploration, a retrosternal mass of 3 by 4 cm with a gross appearance suggestive of thyroid tissue was identified, removed, and referred for pathological examination. The mass was located above the left brachiocephalic vein trunk and anterior to the innominate artery. The thymus remnant (previously identified on the computed tomography scan) was visualised, excised, and sent for a pathology exam (Figure A5).

The patient showed no post-operatory complications, and she was discharged in very good health two days following the intervention (Figure 4). The histological analysis confirmed a benign ectopic mediastinal thyroid, revealing an anisofollicular pattern and thyroid follicles, as well as areas of interstitial haemorrhage with macrophages containing hemosiderin. The thymus nodule consisted of thymus tissue with reactive hyperplasia of the lymphoid tissue and no malignancy. (Also, a small sample of mediastinal fat of 1 by 1 by 0.5 cm was identified) At a macroscopic level, the ectopic thyroid mass was 4 by 3 by 2 cm, while the thymus remnant tissue was 5 by 1 by 1 cm (Figure A6).

### 3.3. Post-Operatory Check-Up

TSH was checked four weeks following thoracic surgery, and it was found to be normal (1.8 µUI/mL). Haemoglobin was also found to be normal (13.4 g/dL; normal ranges between 12 and 15.5 g/dL). Thyroid and parathyroid assays were performed again after three months and showed normal results (Table A1). The cervical scar healed well (Figure A7). An anterior neck ultrasound confirmed stationary aspects of the cervical thyroid, while a normal post-operatory computed tomography aspect was found at the level of the neck and mediastinum (Figure A7).

Long-term endocrine follow-up of the non-functioning adrenal tumour is mandatory. Interestingly, the entire panel of pre-operatory complaints (facial erythema, night sweats, chest pain, and dyspnoea) was completely remitted following surgery. The patient remained in good health for 10 months after surgery.

## 4. Discussion

In this paper, we introduce an unusual case of an adult woman who was confirmed with an ectopic mediastinal thyroid following a COVID-19 infection and required check-up computed tomography. She was surgically managed via an innovative minimally invasive video-assisted procedure. To the best of our knowledge, (according to the data we could find based on the mentioned methods of the literature search), this is the lowest location within the mediastinum of an ectopic thyroid that has been resected through a cervical incision with the preservation of the eutopic gland and involving the use of a Cooper thymectomy device. This resulted in good intra- and post-operatory progress, and sternotomy was not required.

Generally, this distinct endocrine condition (a mediastinal thyroid) remains a rather hidden entity among other more frequent traditional thyroid conditions, such as thyroid nodules at the orthotopic gland [33]. It has an exceptional incidence when compared to other ectopic thyroid tissues (e.g., the lingual thyroid), which, overall, might involve up to 7% of the population according to some data (regardless of the ectopic thyroid site) [34]. This mediastinal ailment may be underdiagnosed, or it might remain asymptomatic across its entire lifespan [35]. Its management is not standard but personalised, and it requires a multidisciplinary team, and its complete removal should be provided by a skilful thoracic surgical team.

While the prior incidental detection of an ectopic mediastinal thyroid has been reported via different imaging assessments [36,37,38,39], we identified only one case with a previous coronavirus infection as the first step that led to the recognition of the mediastinal thyroid [40]. Overall, the panel of conditions, including at the level of the mediastinum, lungs, and pleura, which has been re-shifted in terms of presentation, detection, and management amid the recent pandemic, is more complex [41,42,43,44,45].

The subject presented with a mild clinical picture, which otherwise would not have been investigated (as described from the time before the coronavirus infection). Previous reports of chest pain and dyspnoea have been associated with a mediastinal thyroid [46,47,48]; yet, we found only three papers to address the presence of non-specific night sweats [2,49,50]. This required a differential diagnosis with a pheochromocytoma which was excluded based on an endocrine evaluation (Table A1). Noting the patient’s age, a menopause-related clinical presentation was suspected [51] but not confirmed. As mentioned, these complaints were remitted following thoracic surgery.

Adrenal incidentalomas were more often found amidst a wide area of abdominal findings in COVID-19 patients [52,53,54]. The tumours have an age-related incidence [55,56,57]. We identified only one paper to report the co-presence of an adrenal tumour of 5 cm in a 42-year-old male with an ectopic mediastinal thyroid [58], but no pathogenic connection is known.

We mentioned that our patient delayed any other investigations for one year; hence, a conservative strategy was applied for one year. Yet, during this surveillance, the ectopic thyroid increased by at least 1 cm to the largest diameter (from 3.22 to 4.32 cm). This spontaneous evolution, which was registered, became one more argument for thoracic surgery. Generally, the removal of the mediastinal thyroid represents a choice, and we could only identify three other distinct reports to address surveillance as a first-step strategy [2,59,60], but this is not yet a guideline. This management is distinct from other ectopic thyroid sites due to the specific anatomy of the mediastinal space, including proximity to the heart, large vessels, and airways.

Generally, cervical incision allows the resection of the ectopic mediastinal thyroid in high positions (within the upper part of the superior mediastinum, next to the cervical area). Removal of the eutopic thyroid gland (even without a synchronous indication of a thyroidectomy) has been reported as a sine qua non-step to help this type of surgical procedure gain access to the mediastinal thyroid in lower locations. Additionally, cervicotomy was used in patients with a prior thyroidectomy to access anterior upper mediastinal thyroid masses [39,61,62,63,64,65,66]. Of note, most of the surgeries for a mediastinal thyroid involved other surgical procedures, such as a sternotomy or thoracoscopy [27,39,67,68]. A traditional partial or total sternotomy potentially allows better access to the mediastinal thyroid, but it is associated with a higher rate of complications. A Cooper retractor, in the case presented, allowed better visualisation of the local anatomy and a safe resection with adequate vascular sealing (Figure A2).

To the best of our knowledge, this is the lowest anatomical position within the mediastinum of an ectopic thyroid to be removed via trans-cervical incision with a conservative approach to the eutopic thyroid. This was feasible using a Cooper thymectomy retractor, which also allowed the suspected thymus mass to be removed (it turned out to be thymus hyperplasia). Three other case reports showed the synchronous removal of the thymus with the ectopic mediastinal thyroid, which was not intra-thymic [69,70,71]. The Cooper device has been largely used in various surgical techniques in thoracic surgery; if available, a selected sub-group of candidates might benefit from using it [72,73,74], but traditionally, it belongs to a complex area of thymus tumour-related surgical treatment [75,76,77]. In the presented case, it helped avoid a thyroidectomy since eutopic gland removal was not necessary according to an endocrine assessment (Table 1).

In this case, other highlights are as follows:At the initial computed tomography, a small line of tissue was suspected to be the connective tissue between the orthotopic and ectopic thyroid, but it was not intra-operatory confirmed or after the post-surgery histological exam. This connective tissue facilitates the distinction between ectopic mediastinal tissue and a retrosternal extension of a cervical goitre [79];The eutopic thyroid presented with a micro-nodular pattern, which was not considered a malignancy; that is why a fine needle aspiration was not performed before surgery, and an indication of a concurrent thyroidectomy was not established [80];A small mass of thymus fat was also confirmed according to the post-surgery histological analysis. This was analysed as being in relation to increased total body fat as seen in subjects with a higher body mass index, such as this patient displayed on her first presentation [81,82];While serum thyroglobulin remains a valuable tumour marker upon the resection of thyroid malignancy originating from follicular cells [83], in cases with benign conditions, as seen here, its role is minor.

## 5. Conclusions

This case shows some important aspects. The clinical presentation only became clear to be connected with the ectopic tissue upon the surgical removal of the mediastinal mass and complete remission of the complaints. The incidentaloma (as we may describe the ectopic thyroid based on the initial computed tomography scan) was associated with a second (adrenal) incidentaloma, as often seen in the COVID-19 and early post-COVID-19 era. Minimally invasive cervicotomy associated with eutopic gland conservation and the use of a Cooper thymectomy retractor represent modern aspects in video-assisted thoracic surgery, which provided an excellent outcome and, according to the literature data we searched based on the mentioned methods, involved one of the lowest mediastinal thyroids to be removed by this specific procedure.

## Figures and Tables

**Figure 1 life-14-01374-f001:**
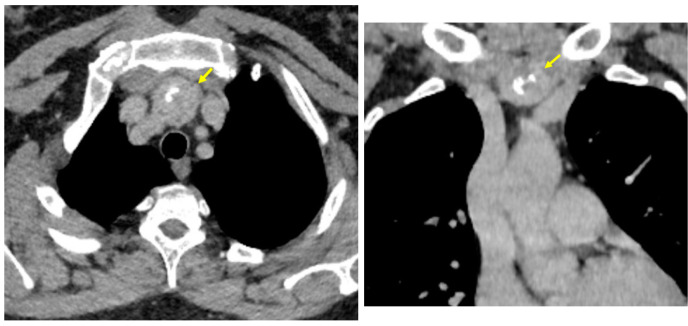
Baseline computed tomography scan. **Upper line**: Intravenous contrast-enhanced computed tomography: a tumour (yellow arrow) of 3.14 by 3.02 by 3.22 cm at the level of the lower part of the upper anterior mediastinum close to the middle mediastinum (**left**: axial plane; **right**: coronal plane). **Lower line**: **Left**: intravenous contrast thoracic computed tomography showing a hypodense nodule in the thymic area of 2.03 by 1.33 cm (axial plane); **Right**: intravenous contrast computed tomography showing an oval, well-shaped tumour on the left adrenal gland of 2.37 by 2.76 by 2.99 cm (adrenal incidentaloma—yellow arrow) located in contact with the splenic artery and vein (anterior), respectively, with upper left kidney pole (caudal); the right adrenal gland had a normal imagery aspect (axial plane).

**Figure 2 life-14-01374-f002:**
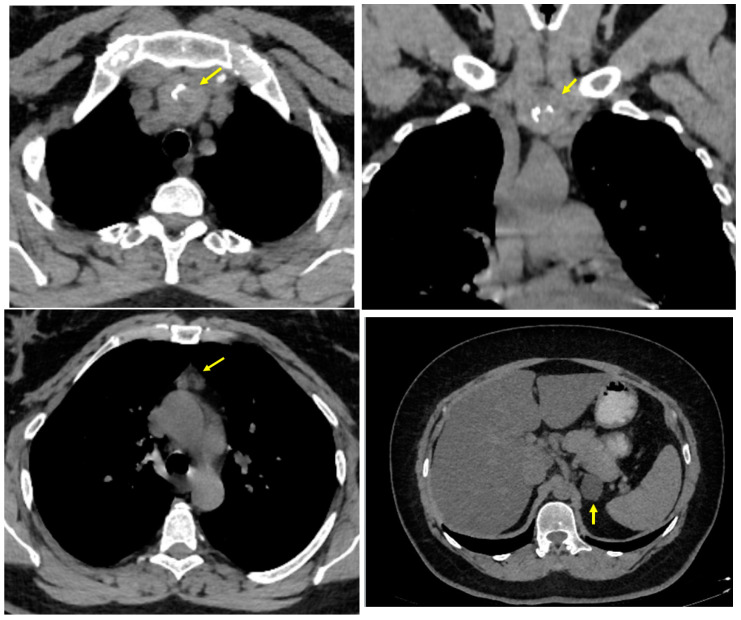
Computed tomography scan after 1-year surveillance. **Upper line**: Non-enhanced thoracic computed tomography scan: solid nodule (yellow arrow) with heterogeneous aspect and micro- and macro-calcifications, with similar density to the thyroid parenchyma (63 HU) in the lower part of the upper anterior mediastinum (3.27 by 3.22 by 4.31 cm; the lower pole comes in close contact with subclavian arteries (bilaterally) (**left**: axial plane; **right**: coronal plane). **Lower line**: **Left**: Non-contrast enhanced computed tomography scan: a hypodense nodule (yellow arrow) of 1.55 by 1.74 cm at the level of thymus area (axial plane); **Right**: Non-enhanced abdominal computed tomography scan: oval, hypodense, well-shaped tumour mass (yellow arrow) of 2.4 by 2.7 by 3 cm (−3 HU) in the body of the left adrenal gland (adrenal incidentaloma) (axial plane).

**Figure 3 life-14-01374-f003:**
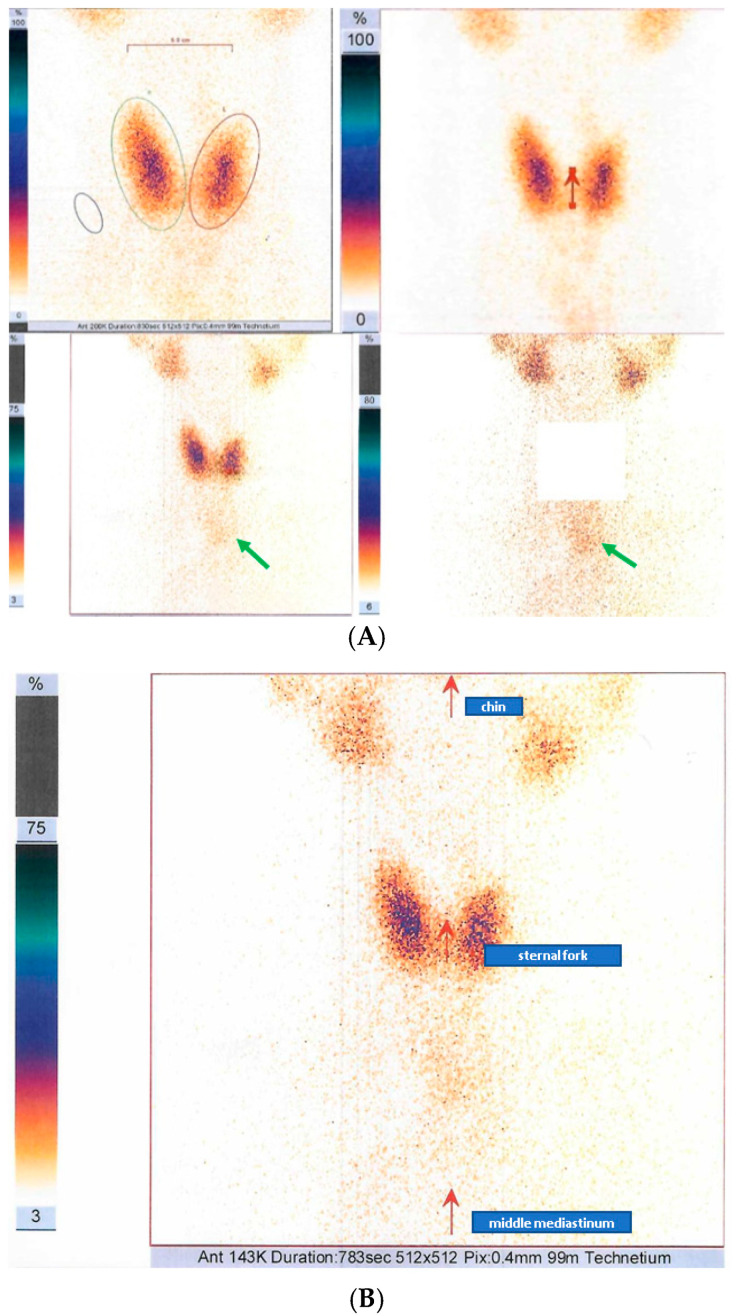
99m-Tc pertechnetate scintigraphy (259 MBq). (**A**) Inhomogeneous area of tracer uptake at the level of median line at the upper mediastinum without contact with the thyroid gland, suggesting an ectopic mediastinal thyroid (green arrow). (**B**) Anatomical landmarks (red arrows) in 99m-Tc scintigraphy; the ectopic thyroid was situated next to the middle mediastinum.

**Figure 4 life-14-01374-f004:**
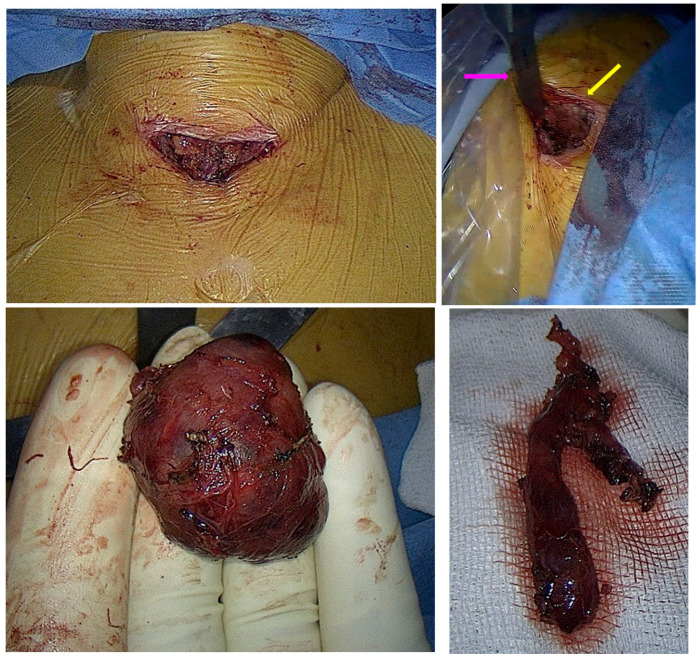
Intra-operatory aspects. **Upper line**: Intra-operatory captures: (**left**) cervical incision; (**right**) Cooper thymectomy retractor retrosternal blade in place (pink arrow); cervical incision (yellow arrow). **Lower line**: Gross specimen of the removed masses: (**left**) ectopic thyroid gross specimen; (**right**) thymus gross specimen.

**Table 1 life-14-01374-t001:** Unusual aspects of the presentation and particular highlights of the management in the reported case amid similar reported data according to our literature review [2,39,40,49,50,58,59,60,61,62,63,64,65,66,69,71,78].

Unusual Findings in this Case	Similar Reports—First Author	Reference
Detection upon prior COVID-19 infection	Melinte A.	[40]
Pre-surgery presentation with intermittent night sweats (and remission following ectopic thyroid removal)	Motlaghzadeh Y.	[2]
	Guimarães MJ.	[49]
	Barker TA.	[50]
Co-presence of a unilateral adrenal incidentaloma	Kola E.	[58]
Conservative management	Motlaghzadeh Y.	[2]
	Hummel J.	[59]
	Agrawal K.	[60]
Trans-cervical incision for removal of the mediastinal thyroid	Aziz J.	[39]
	Imai T.	[61]
	Metere A.	[62]
	Scognamillo F.	[63]
	Walz PC.	[64]
	Mace AD.	[65]
	Uchida N.	[66]
Synchronous thymus mass removal	Gao M.	[69]
	Muzurović E.	[71]
	Kamaleshwaran KK.	[78]

## Data Availability

Other data are available upon reasonable request.

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
