# Peer review of "Ectopic Mediastinal Thyroid: A Crossroad Between a Multi-Layered Endocrine Perspective and a Contemporary Approach in Thoracic Surgery"

_life, 2024, doi:10.3390/life14111374_

Round 1

Reviewer 1 Report

Comments and Suggestions for Authors

Dear Authors,

Thank you for your submission. The clinical case you present is both intriguing and of interest to the readership. However, I believe that the manuscript would benefit from a few adjustments to enhance its readability and engagement.

The abstract, in my opinion, should be rewritten to be more concise, direct, and streamlined. It would be beneficial to focus on a few key points that immediately capture the reader's attention.

In the "Introduction" section (lines 84–118), I suggest being more direct in addressing the specific "gap" in the literature that your clinical case helps to clarify. It’s important to engage the reader from the start before presenting the aims, which I recommend streamlining as well.

The second part of the manuscript appears to be well-structured.

I believe these adjustments will significantly improve the manuscript’s clarity and impact.

Comments on the Quality of English Language

The English could be made smoother, and I believe it would benefit from a review by a native speaker to ensure the text is more concise and clearly conveys the authors' intended message.

Author Response

Response to Review 1 Comments

Dear Reviewer,

Thank you very much for your time and your effort to review our manuscript.

We are very grateful for providing your valuable feedback on the article.

Here is our response and related amendment that has been made in the manuscript according to your review (marked in yellow color).

Dear Authors,

Thank you for your submission. The clinical case you present is both intriguing and of interest to the readership.

Thank you very much. We really appreciate it!

However, I believe that the manuscript would benefit from a few adjustments to enhance its readability and engagement.

Thank you very much. We addressed them as follows:

The abstract, in my opinion, should be rewritten to be more concise, direct, and streamlined. It would be beneficial to focus on a few key points that immediately capture the reader's attention.

Thank you very much. According to your recommendation, we rephrased and corrected the abstract and focused on the main key findings. Thank you

In the "Introduction" section (lines 84–118), I suggest being more direct in addressing the specific "gap" in the literature that your clinical case helps to clarify. It’s important to engage the reader from the start before presenting the aims, which I recommend streamlining as well.

Thank you very much. According to your recommendation, we rephrased and corrected the mentioned sub-sections from Introduction and Objective. Thank you

The second part of the manuscript appears to be well-structured.

Thank you very much.

I believe these adjustments will significantly improve the manuscript’s clarity and impact.

We agree. Thank you

Comments on the Quality of English Language: The English could be made smoother, and I believe it would benefit from a review by a native speaker to ensure the text is more concise and clearly conveys the authors' intended message.

Thank you. We revised the text and the English language.

Thank you very much.

Reviewer 2 Report

Comments and Suggestions for Authors

Authors describe a case report of ectopic thyroid tissue generating night sweats symptoms that have dissapeared after  surgery. The association with  adrenal incidentaloma seems to be related to CTs realised more often in the context of COVID pandemics

Author Response

Response to Review 2 Comments

Dear Reviewer,

Thank you very much for your time and your effort to review our manuscript.

We are very grateful for your insightful comments and observations, also, for providing your valuable feedback on the article.

Here is a point-by-point response and related amendments that have been made in the manuscript according to your review (marked in yellow colour).

Authors describe a case report of ectopic thyroid tissue generating night sweats symptoms that have disappeared after surgery. The association with adrenal incidentaloma seems to be related to CTs realized more often in the context of COVID pandemics.

Thank you very much. Indeed, one of the particular insights in this case relates to the COVID-19 pandemic and the consecutive incidental finding amid serial CT scans. Also, the presence of night sweats is highly non-specific, but the post-operatory remission confirmed the connection with the mediastinal thyroid. Thank you

Thank you very much.

Reviewer 3 Report

Comments and Suggestions for Authors

The reviewer thanks the authors for submission of this interesting case report. I offer my comments and suggestions with the highest respect and in recognition of the teams outstanding clinical care of their patient 

Comments on the Quality of English Language

The manuscript should be significantly shortened and refocused on the core novel clinical issues. There are problems with the English language, syntax and word use that pose a distraction and challenge but these can be rectified with a thorough editing. 

Author Response

Response to Review 3 Comments

Dear Reviewer,

Thank you very much for your time and your effort to review our manuscript.

We are very grateful for your insightful comments and observations, also, for providing your valuable feedback on the article.

Here is a point-by-point response and related amendments that have been made in the manuscript according to your review (marked in yellow colour).

The reviewer  thanks the authors for submission of this interesting case report. I offer my comments and suggestions with the highest respect and in recognition of the teams outstanding clinical care of their patient 

Thank you very much. We really appreciate it!

General Comments on review

This reviewer sends highest compliments to the authors of this case report with also reported additional challenges and complications. As an academic medical Endocrinologist, I consider their care rendered to their patient as outstanding and equal to anywhere in the world.

In general case reports serve a crucial role in evidence-based medicine as they can provide unique insights, challenges in diagnosis and approaches to therapy. They are critical to both educate medical learners such as students, residents and fellows but also practitioners in medicine and surgery. Case reports in general should be focused on the primary “teachable” event and be at best highly novel by providing some interesting insight to potentially change the approach to diagnosis and therapy.

This case report of the discovery of a true ectopic thyroid mass within the mediastinum essentially discovered as an “incidentaloma” holds some interest. This lesion is very rare and often poses challenges to the best of surgical and medical experts. They also report the discovery of an adrenal incidentaloma and carried out and a standard process to investigate adrenal functioning.

Thank you very much.

Unfortunately, this manuscript has many challenges which I will detail below.

Thank you very much. We addressed them as follows:

However, before offering my criticisms, I would like to offer that there is novelty in this case report which I believe is the greatest strength. This obviously outstanding- surgical/ medical team took a novel surgical approach to resection of the mediastinal mass with complete success and preserved the thyroid gland. In addition, there journey towards diagnosis and surgical success provide an excellent instructional journey for Pulmonologists, Endocrinologists and surgeons with a mediastinal mass. It is the reviewer’s opinion that these are the key important clinical events and interventions that should be the focus of the paper along with the description of the rarity of this “incidentaloma” and the approach to diagnosis. Also, the inclusion of the description in line 415-420 of the decision to elect this surgical approach preserving the eutopic thyroid is again a strength.

Thank you very much. This work was indeed a merge between the thoracic surgery team and the endocrine team and we appreciate pointing out this multidisciplinary perspective.

On reviewing the references, the following appears to be a particular outstanding case series report from Stanford cited by the authors and may provide a template for improving the manuscript:

Yasaman, Motlaghzadeh., Shannon, Nesbit., H., Henry, Guo., Eric, Yang., Kaniksha, Desai., Natalie, S., Lui. (2022). 6. Surgical resection of mediastinal ectopic thyroid tissue: a case series. Journal of Thoracic Disease, doi: 10.21037/jtd-22-479

Thank you very much. We followed your useful recommendation and added it. Thank you

There are so many other distracting issues that are of course important to the care of the patient but don’t really add to the critical.

Thank you very much. We addressed them point by point.

Quality of scientific writing. In general, the paper has many issues with the description of events and their use of unconventional terms and words. For example, the use of “bizarre” in line 38 and 84 is more jargon and should be replaced with more acceptable terms. There are also numerous other sentences with poor syntax that should be revised or eliminated and as such a very thorough editorial editing is needed throughout.

Thank you very much. According to your recommendation, we revised the manuscript, corrected the English language, and removed the term “bizarre” from the abstract and the main text. Thank you

Extemporaneous-additional events of low academic value- This is perhaps the greatest distraction of the manuscript. The authors describe and discuss many additional clinical discoveries and events that are distracting from the key strength of the paper and as such lower the quality of this report. The following are items I would eliminate or greatly minimize.

Although discovery of an adrenal “incidentaloma” is of direct clinical importance to the patient there is little academic value related to this report. A simple sentence or two reporting the endocrine work was negative for a functional tumor and no growth in the lesion and therefore no indication for surgery. Also add something to the effect there was no evidence of a connection with the ectopic thyroid lesion would suffice. Describing in detail the work up and other as stated in text and discussion adds little to the importance of the case.

Thank you very much. According to your recommendation, we reduced the data with respect to the adrenal incidentaloma in the main text, including at Discussion and removed the mentioned subsections. Thank you

Section 4.1- This section is not well written in terms of understanding the relevance. It could be greatly condensed with the essential novel insight provided or just fold into the general description of the pathway to discovery.

Thank you very much. According to your recommendation, we removed this micro-section (4.1.) from Discussion. Thank you

Description of the eutopic thyroid and issues related to the assessment and possible biopsy could be summarized in one sentence.

Thank you very much. According to your recommendation, we summarized it in one sentence. Of note, this issue represented a crucial aspect with regard to deciding to conserve the eutopic thyroid and not perform a thyroidectomy during thoracic surgery (eutopic thyroid removal allows a much easier surgical access to the mediastinal thyroid tissue during thoracic surgery). We respectfully consider being essential this particular aspect to be mentioned at least in one sentence. Thank you

Line 425-428 The discussion of thyroglobulin and thyroid antibodies is not of added interest as this is well known by endocrinologists and the findings could be shortened.

Thank you very much. According to your recommendation, we summarized it in one sentence.

Line 429-436. This description again is largely unrelated and as such distracts from the important core issue and could be eliminated.

Thank you very much. According to your recommendation, we eliminated those lines.

Section 4.4- questioning the “conservative” approach- It would be best to describe the related study experiences with a “wait and see” approach and compare to your outcomes to provide a more general insight. You have listed the published papers and as such you could provide a table of their clinical descriptions and decision for surgery.

Thank you very much. In the specific field of the mediastinal ectopic thyroid (and not generally in the field of any type/site of ectopic thyroid tissue since mediastinal thyroid is different because mediastinum poses distinct anatomic features), we are not aware of any specific study with regard to “wait and see” approach (longitudinal cohorts), and the natural history of the condition remains largely unknown. Of note, there is no guideline, neither a large study have been published, only limited series and case reports. In this reports, the decision was either conservative or surgical. Since the level of statistical evidence remains very low, no general conclusion can be established at this point, thus the importance of such case as seen here whereas the conservative approach was followed by a surgical management with a post-operatory histological confirmation. In addition, we respectfully mention that the cited papers in Table 1 represent connected papers with regard to the key findings (particular aspects) of this case (co-presence of an adrenal incidentaloma, co-presence of a thymus mass, detection after COVID-19 infection, etc.), and not a literature review of surgery candidates, this being out of the scope with respect to this case report. Thank you very much

Comments on the Quality of English Language

The manuscript should be significantly shortened and refocused on the core novel clinical issues. There are problems with the English language, syntax and word use that pose a distraction and challenge but these can be rectified with a thorough editing. 

Thank you very much. According to your recommendation, we revised the text, refocused on the main findings, reduced the length of the abstract and manuscript, rephrased the manuscript, reorganized the figures, and corrected the English language. Thank you

In summary there is clinical and educational value to this case report if the authors would focus on the strengths as noted above and minimize or eliminate the distractions. They are to be congratulated on their exceptional clinical care.  

Thank you very much.

Reviewer 4 Report

Comments and Suggestions for Authors

It is just a case study with a poorly performed systematic review. As a consequence, it is not very informative for the reader. The link between ectopic thyroid tissue was not documented.

The case should be presented in line with CARE guidliness.

The systematic review should summarise at least the location of the tissue.

The abstract should be more consistent and not replicate the introduction and result sections. 

In my opinion, the authors should provide a systematic review based on a literature search, which should be clearly described in the methodology.

There is 729 records in PubMed for ectopic thyroid. (COVID-19 is redundant). Some systematic review should be added. 

The presentation of the case should present the important data, while redundant should be removed (e.g. 'Biochemical assays showed normal liver enzymes (in the mean time she normalised her body mass index) with a mild elevation of the uric acid) and similar.

There are too many figures. Please combine them in time points. (Figer 1 and 2  and 3 as one). There is nothing unusual in adrenal incidentaloma - a single picture is enough. Figure 4 as supplementary. Figures 5-7 as one. For figures 9-12, select four pictures and create one figure. Delete 13 or put it as supplementary.

Please specify what is so unusual. Please explain the link between ectopic thyroid and symptoms - why? The co-existence with adrenal incidentaloma is by chance?

Delete incidentaloma scenario upon COVID-19 infection it is redundant and not informative.

Author Response

Response to Review 4 Comments

Dear Reviewer,

Thank you very much for your time and your effort to review our manuscript.

We are very grateful for your insightful comments and observations, also, for providing your valuable feedback on the article.

Here is a point-by-point response and related amendments that have been made in the manuscript according to your review (marked in yellow colour).

It is just a case study with a poorly performed systematic review. As a consequence, it is not very informative for the reader. The link between ectopic thyroid tissue was not documented.

Thank you very much.

We respectfully mention the fact that this is a case report, a systematic review being out of the scope of the present work.

We searched the literature according to the mentioned methods with regard to the key findings of this case (particular aspects) and analysed these specific data as presented in Discussion section.  

Notably, with respect to the mediastinal ectopic thyroid: this is distinct from other types/sites of ectopic thyroid (which are dominated by the lingual site in 90% of cases). The mediastinum poses specific anatomic traits which require a specific management, including in terms of diagnosis tools (such as biopsy, etc.) or surgical removal.  Under these circumstances, this paper focused only on the mediastinal site which is extremely rare. Due to this rarity, no systematic review with regard to mediastinal ectopic thyroid is available so far, neither this is the topic of a specific guideline, thus the importance of addressing this issue on a case by case strategy.

Also, noting these particular aspects of the mediastinum and associated surgical issues, we applied the manuscript to a Special Issue dedicated to thoracic surgery which does not regard other non-mediastinal sites of the ectopic thyroid as, for instance, often found in the lingual thyroid.

Thank you

The case should be presented in line with CARE guidliness.

Thank you very much. According to your recommendation, we provided the capture of the case report timeline that is presented according to CARE guidelines (Figure S). Thank you for this interesting recommendation.

The systematic review should summarise at least the location of the tissue.

Thank you very much.

We searched “ectopic mediastinal thyroid” (we already mentioned why mediastinal site cannot be assimilated into a general management of any ectopic thyroid location) and found 332 records in PubMed, as shown below:

Then we restricted to “systematic review” and found only 3 papers, as follows:

Two of these 3 papers focused on ectopic ACTH syndrome and another one focused on using SPECT/PET/CT for thyroid cancer in ectopic tissue.

To our aware, a systematic review in the specific topic of an ectopic mediastinal thyroid is not feasible at this point since there are only case series and small studies or studies which incorporated a small number of patients with this distinct location (as opposite to other ectopic thyroid sites), hence, the studies design largely vary and they cannot be submitted for a homogenous statistical analysis.

In addition, to our knowledge, the largest analysis on prior published data/reports with regard to the ectopic mediastinal thyroid has been published in 2024 in another MDPI journal (https://pubmed.ncbi.nlm.nih.gov/38791947/) and we do not intend to replicate that article, but to add to the limited original data/reports that have been published so far in the literature concerning this specific topic of the ectopic mediastinal thyroid.

Thank you

The abstract should be more consistent and not replicate the introduction and result sections. 

Thank you very much.

According to your recommendation, we rephrased and corrected the abstract and focused on the main key findings. Thank you

In my opinion, the authors should provide a systematic review based on a literature search, which should be clearly described in the methodology.

Thank you very much. We already addressed this issue. Thank you

There is 729 records in PubMed for ectopic thyroid. (COVID-19 is redundant). Some systematic review should be added. 

Thank you very much. We already addressed this issue. Thank you

The presentation of the case should present the important data, while redundant should be removed (e.g. 'Biochemical assays showed normal liver enzymes (in the meantime she normalized her body mass index) with a mild elevation of the uric acid) and similar.

Thank you very much.

According to your recommendation, we removed this paragraph and revisited the data of the case. We respectfully mention that the case is a presentation from a surgical, but, also, an endocrine perspective since ectopic thyroid at mediastinum level concerns a multidisciplinary team. Thank you

There are too many figures. Please combine them in time points. (Figer 1 and 2  and 3 as one).

Thank you very much.

According to your recommendation, we combined Figure 1, 2, and 3.

There is nothing unusual in adrenal incidentaloma - a single picture is enough.

Thank you very much.

According to your recommendation, we removed Figures 3B and 7B (the second capture for the adrenal incidentalomas at baseline and during follow-up).

Figure 4 as supplementary. Figures 5-7 as one. For figures 9-12, select four pictures and create one figure. Delete 13 or put it as supplementary.

Thank you very much.

According to your recommendation, we removed, combined or moved those figures are supplementary. Thank you

Please specify what is so unusual.

Thank you very much. This case report includes the following unusual aspects:

  • the discovery of a true ectopic thyroid mass within the mediastinum (essentially discovered as an incidentaloma) with a mildly symptomatic clinical picture that was not suggestive to provide a clinical index of suspicion for this disease
  • This lesion is very rare and often poses challenges to the best of surgical and medical experts.
  • The thoracic surgical approach included a cervicotomy, not a (traditional) sternotomy (minimally invasive approach with best recovery results)
  • The thoracic surgical approach also allowed the conservation of the eutopic thyroid gland considering the fact that the easier access to the mediastinum mass is by firstly providing a thyroidectomy
  • Post-operatory remission of the clinical picture confirmed the connection with the mediastinal thyroid
  • The co-presence of an adrenal incidentaloma, and a thymus mass makes the case even rarer.

Thank you

Please explain the link between ectopic thyroid and symptoms - why?

Thank you very much. All four symptoms had been reported in other mediastinal masses/tumours/neoplasia, too (other than ectopic mediastinal thyroid). Dyspnoea and chest pain are related to the anatomical presence of a mass/tumour at the level of mediastinum which is a limited space with potential compressive effects on lung-related expansion amid breathing, including an exacerbation during physical exercise. Facial erythema is related to the compression on the large vessels situated at the level of mediastinum (some captures provided in the article pinpointed these anatomical aspects) while a clear venous obstruction-related Pemberton sign was not confirmed in this patient. Sweats may underline multifactorial components: they may be related to a transitory adrenaline-associated head production due to above mentioned issues, or a transitory increase of the metabolic rate at the level of ectopic thyroid tissue with regard to the local thyroid hormones production (which is not translated into a higher level of their blood hormones during current assays) or local cytokines over-production that has been described in non-thyroid masses (https://www.ncbi.nlm.nih.gov/books/NBK546608/). Thank you

The co-existence with adrenal incidentaloma is by chance?

Thank you very much. Indeed, we mentioned that so far no pathogenic connection had been established to the ectopic thyroid and only one prior report identified a similar finding. Modern medical era showed that the accidental detection has been accelerated by a larger number of imaging scans, including CT scans, that have been done amid recent COVID-19 pandemic and, hence, it led to the detection of many prior unknown or unconfirmed ailments that might have been completely asymptomatic or mildly symptomatic as seen in this case. Thank you

Delete incidentaloma scenario upon COVID-19 infection it is redundant and not informative.

Thank you very much.

According to your recommendation, we removed this micro-section (4.1.) from Discussion. Thank you

Thank you very much.

Round 2

Reviewer 1 Report

Comments and Suggestions for Authors

Dear Authors,

Thank you for addressing the revisions. The changes you have made have significantly improved the clarity of the manuscript, making it more accessible and easier to read.

I have just one suggestion regarding the figures. In line with the journal's guidelines (see "All Figures, Schemes and Tables should have a short explanatory title and caption," available at https://www.mdpi.com/journal/life/instructions#figures), I recommend including a title next to the figure number (e.g., Figure 1. "(Title)"). The caption can then be placed directly below the title. This adjustment will ensure consistency and adherence to the journal’s formatting requirements.

Thank you

Author Response

Response to Review 1 Comments

Dear Reviewer,

Thank you very much for your time and your effort to review our manuscript.

We are very grateful for providing your valuable feedback on the article.

Here is our response and related amendment that has been made in the manuscript according to your review (marked in yellow green).

Dear Authors,

Thank you for addressing the revisions. The changes you have made have significantly improved the clarity of the manuscript, making it more accessible and easier to read.

Thank you very much.

I have just one suggestion regarding the figures. In line with the journal's guidelines (see "All Figures, Schemes and Tables should have a short explanatory title and caption," available at https://www.mdpi.com/journal/life/instructions#figures), I recommend including a title next to the figure number (e.g., Figure 1. "(Title)"). The caption can then be placed directly below the title. This adjustment will ensure consistency and adherence to the journal’s formatting requirements.

Thank you

Thank you very much. According to your recommendation, we included the title for each figure next to the number. Thank you

Reviewer 3 Report

Comments and Suggestions for Authors

Thank you for your excellent revisions

Comments on the Quality of English Language

Improvement overall 

Author Response

Response to Review 3 Comments

Dear Reviewer,

Thank you very much for your time and your effort to review our manuscript.

We are very grateful for your insightful comments and observations, also, for providing your valuable feedback on the article.

Thank you for your excellent revisions

Thank you very much.

Comments on the Quality of English Language: Improvement overall 

Thank you very much.

Reviewer 4 Report

Comments and Suggestions for Authors

The paper was improved. No further comments.

Author Response

Response to Review 4 Comments

Dear Reviewer,

Thank you very much for your time and your effort to review our manuscript.

We are very grateful for your insightful comments and observations, also, for providing your valuable feedback on the article.

Comments and Suggestions for Authors: The paper was improved. No further comments.

Thank you very much.
